Microsatellite frequencies vary with body mass and body temperature in mammals, suggesting correlated variation in mutation rate

Amos William 1 w.amos@zoo.cam.ac.uk
Filipe Laura N.S. 2
1 Department of Zoology , Cambridge , UK
2 Hills Road Sixth Form College , Cambridge , UK
Ross-Ibarra Jeffrey
Electronic publication date: 2014 Nov 6
Publication date: 2014
Volume: 2
Electronic Location ID: e663
Received 2014 Jun 22; Accepted 2014 Oct 21
Copyright: © 2014 Amos and Filipe
Copyright year: 2014
Copyright holder: Amos and Filipe
License: This is an open access article distributed under the terms of the Creative Commons Attribution License, which permits unrestricted use, distribution, reproduction and adaptation in any medium and for any purpose provided that it is properly attributed. For attribution, the original author(s), title, publication source (PeerJ) and either DOI or URL of the article must be cited.
License URL: https://creativecommons.org/licenses/by/4.0/

Keywords: Microsatellite, Heterozygosity, Mammal, Evolution, Mutation rate, Phenotype

Funding: The authors declare there was no funding for this work.

==============================
Substitution rate is often found to correlate with life history traits such as body mass, a predictor of population size and longevity, and body temperature. The underlying mechanism is unclear but most models invoke either natural selection or factors such as generation length that change the number of mutation opportunities per unit time. Here we use published genome sequences from 69 mammals to ask whether life history traits impact another form of genetic mutation, the high rates of predominantly neutral slippage in microsatellites. We find that the length-frequency distributions of three common dinucleotide motifs differ greatly between even closely related species. These frequency differences correlate with body mass and body temperature and can be used to predict the phenotype of an unknown species. Importantly, different length microsatellites show complicated patterns of excess and deficit that cannot be explained by a simple model where species with short generation lengths have experienced more mutations. Instead, the patterns probably require changes in mutation rate that impact alleles of different length to different extents. Body temperature plausibly influences mutation rate by modulating the propensity for slippage. Existing hypotheses struggle to account for a link between body mass and mutation rate. However, body mass correlates inversely with population size, which in turn predicts heterozygosity. We suggest that heterozygote instability, HI, the idea that heterozygous sites show increased mutability, could provide a plausible link between body mass and mutation rate.

Introduction

Classical studies of molecular evolution often refer to the ‘molecular clock’ (Zuckerkandl & Pauling, 1962; Margoliash, 1963), the notion that genetic mutations occur randomly and at constant rate such that the degree of divergence seen between species tends to be proportional to geological time. However, even from its early days, the constancy of the clock has been questioned (Fitch, 1983). Among many deviations, one of the most intriguing is the way the rate of molecular evolution appears to covary with life history traits (Martin & Palumbi, 1993; Smith & Donoghue, 2008; Lanfaer, Welch & Bromham, 2010; Bromham, 2011). A good example is body mass in mammals, where larger species evolve more slowly than smaller species (Bromham, Rambaut & Harvey, 1996; Lartillot & Poujol, 2011). The most widely accepted explanation for this involves a link to generation length, but fit to a simple model is poor, in that a doubling of generation time on average results in much less than a halving of evolutionary rate (Welch, Bininda-Emonds & Bromham, 2008).

Despite many studies exploring the relationship between life history traits and evolutionary rate, the overwhelming majority have focused on either mitochondrial DNA or nuclear ‘genes’. Few, if any, have looked explicitly at the fastest evolving components of the genome such as short tandem repeats (STR), yet it is here that the largest effects might be expected. Microsatellites form an important class of STR with repeats one to six nucleotides in length and occur abundantly in higher organisms (Bruford & Wayne, 1993). Most mutations involve molecular slippage causing the gain or loss of repeat units (Schlötterer & Tautz, 1992), often with bias favouring contraction or expansion (Dermitzakis et al., 1998; Ellegren, 2000; Xu et al., 2000). Mutation rate increases with repeat number in a strongly non-linear fashion (Weber, 1990; Kelkar et al., 2008). Maximum repeat number of a given microsatellite motif varies between species (Di Rienzo et al., 1994), probably due to a thermally-influenced upper length boundary (Amos & Clarke, 2008).

Microsatellites seem to be ‘born’ when point mutations by chance create sufficient consecutive repeats for slippage to occur (Messier, Li & Stewart, 1996), persist for a finite time and eventually ‘die’ (Taylor, Durkin & Breden, 1999). The process of death is poorly understood and may well vary between loci, some being deleted while others degenerate through substitutions within the repeat array. However, regardless of the exact details, the relative frequency of any given length of microsatellite in the genome should tend to reflect the relative time spent by an average microsatellite during its lifetime at that length. Short microsatellites are common both because all microsatellites are short at some point and because their low mutation rate causes them to linger at this length. Conversely, long microsatellites are rare both because many loci never become long and because, once a locus becomes long, the much higher mutation rate causes it rapidly either to shrink or to ‘die’. Consequently, the length-frequency distribution of microsatellites in the genome can be viewed as a form of standing wave with microsatellite birth acting as a source and microsatellite death a sink. The exact shape of this wave will reflect a complex interplay between the rates of birth and death and the relationships between repeat number, mutation rate and mutation bias.

Differences between species in the shape of the standing wave may allow a distinction to be made between two key hypotheses concerning life history traits. In the generation length hypothesis, variation in mutation rate arises from differences in the number of mutation opportunities per unit time. We call these mutation number models. In contrast, factors such as metabolic rate or body temperature (Tuntiwechapikul & Satazar, 2002) may impact mutation rate directly by changing the probability that a mutation occurs in any given round of DNA replication. We call these mutation rate models. For base substitutions, the two models yield identical predictions in terms of sequence divergence. However, the same does not necessarily hold for microsatellites.

If the system is at or very close to equilibrium the shape of the standing wave will be stable and should not change in response to more or fewer mutations occurring when generation length shortens or lengthens respectively. The shape of the standing wave should only change in response to changes in the parameters that influence its shape, specifically if the relative mutability of different length microsatellites changes or if the relationship between mutation bias and repeat number changes. For example, if longer microsatellites become relatively more mutable, an average microsatellite will spend less time ‘long’, and a new equilibrium wave will be created in which the frequency of long microsatellites is reduced. Such changes should not occur under mutation number models but are a possible (though not necessary) consequence of mutation rate models.

To help understand the relationship between life history traits and mutation rate we used mammalian genome sequences to determine length-frequency profiles for three common microsatellite motifs. We then asked: (a) whether these profiles differ between species and (b) whether any differences relate to two key phenotypes, body temperature or body mass (Gillooly et al., 2005).

Materials and Methods

Data

We downloaded 69 complete or largely complete mammalian genomes from the NCBI and UCSC websites (http://www.ncbi.nlm.nih.gov/, http://genome.ucsc.edu/), some as separate assembled chromosomes and others as bulk files of larger, unplaced contigs. Given the different formats, and for complete control, we counted microsatellites using custom C++ scripts written by WA, focusing on three motifs: ‘AC’, ‘AT’ and ‘AG’ (see Fig. S1 for annotated example script). Together with their reverses (‘CA’, ‘GA’, ‘TA’), complements (‘TG’, ‘TC’) and their reverse complements (‘GT’, ‘CT’) these embrace all possible dinucleotides with the exception of ‘CG’ and ‘GC’, which are extremely rare due to the high mutability of CpG (Kelkar et al., 2008). In fact, we did also look at (CG) repeats but their general rarity precluded a meaningful analysis: tracts carrying nine or more repeats were absent from at least some species and human (CG)3 is 30 fold rarer than (AT)3. A lower limit of four repeats was set to reflect the shortest length at which slippage is likely important. An upper limit of 30 repeats was set because longer tracts are extremely scarce in some species, probably due to a combination of the known dependence on body temperature and biases arising from differences between sequencing platforms. We also looked at three triplet repeats, ‘ATT’, ‘AGG’ and ‘AAC’. Substitutions within the repeat tract occur at an appreciable rate and act to reduce the rate of slippage (Jin et al., 1996).

Since a minority of tracts are interrupted and interrupted tracts seem to mutate approximately as if they comprise just the longest pure tract (Weber, 1990), we began by counting only pure repeat tracts. No minimum separation was enforced between tracts, such that (AC)10AT(AC)4 would count one (AC)10 and one (AC)4: the longer tract would count anyway while the shorter tract would add negligibly to the very large number of isolated (AC)4 tracts. To explore the possible impact of interruption mutations we then repeated the analysis using tracts of the form (XY)aZZ(XY )b, where XY is the target motif, ZZ represents any pair of bases that is not the target motif. We required a and b both to be greater than one and that either or both were greater than two.

Statistical analysis

All analyses were conducted using R 2.15.2 (http://cran.r-project.org/). To control for shared ancestry, we used the method of phylogenetically independent contrasts (PICs) (Felsenstein, 1985; Purvis & Rambaut, 1995), implemented in the package ‘ape’ (Paradis, Claude & Strimmer, 2004). Calculation of PICs requires a phylogeny with associated branch lengths (Fig. S1). Wherever possible, including all deeper nodes, dates were read from Fig. 1 in Meredith et al. (2011). Dates for other nodes were extracted from individual group phylogenies (Fernandez & Vrba, 2005; Osada et al., 2008; Agnarsson et al., 2011; Perelman et al., 2011; Fabre et al., 2012) (for the resulting phylogeny, see Fig. S2).

Figure 1 Variation in relative frequency of different length AC microsatellites in six higher primates.

The six higher primates are: human (HAS, black), bonobo (PPA, blue), chimpanzee (PTR, red), gorilla (GGO, pink), orang-utan (PAB, green) and gibbon (NLE, yellow). All frequencies are expressed as natural logarithms and individual data points are then calculated as deviations from the group mean. A value of one thus indicates a frequency 2.7 times the geometric mean of the six species.

To test whether microsatellite frequency differences correlate with life history characteristics we focused on two phenotypes, body temperature and sex-averaged adult body mass. Body temperatures vary with factors such as exercise and digestion, and can fall dramatically during hibernation or torpor. We therefore sought values for the ‘normal’ active temperature, and for consistency took as many measurements as possible from one large study by Clarke & Rothery (2008) which also gives body masses (Table S1). For four temperatures and one mass we were unable to find reliable values and had to use educated guesses based on closely related species (for details, see Table S2). For each repeat number (range four to 30) in each of the three dinucleotide motifs (AC, AG, AT) we fitted multiple linear regressions with loge (frequency) as the response and body temperature (°C), log10 (body mass, Kg) and %GC content as predictors. %GC content was included to control for the possibility that this influences microsatellite frequencies and has a small but marginally significant impact on the frequencies of the very shortest microsatellites (repeat number <10). All variables were initially (see below) included as 68 PICs. Once adequate model fit had been achieved, significance was determined using the command ‘anova’ to compare models with and without the term of interest being fitted. Thus, if the full model is written F ∼ P1 + P2 + GC (F, log frequency; P1, phenotype1; P2, phenotype 2; GC = %GC); the significance of P1 would be assessed by comparing the full model with the model F ∼ P2 + GC.

Model fit was explored using the command ‘plot’, with particular attention given to models yielding high significance. In most cases the residuals were reasonably normally distributed and there were no major outliers or points with high leverage (see Fig. S3 for typical diagnostic plots). However, in some of the PIC models one data point, that corresponding to the deep node separating rodents and primates, appeared as a big outlier across all motifs and was excluded from all analyses. Two other deep nodes, primate-Eulipotyphla and Eulipotyphla-carnivore, were medium outliers for some motifs but others and were excluded from all models of the motifs where they were a problem.

Results

(a) Do closely related species differ in their microsatellite frequencies?

We began by asking whether closely related species exhibit potentially informative differences in their microsatellite frequencies, based on the well-curated genomes of six higher primates (human, chimpanzee, bonobo, gorilla, orang-utan and gibbon). The frequency of any given species—microsatellite length—motif combination, Fijk, was expressed on a log scale: Fijk=logemijk+1ni

where mijk is the raw count of that microsatellite motif k, length j in species i and ni is the number of nucleotide bases assayed in the genome of species i. To create profiles depicting the relative frequencies of each microsatellite in each species across the range of lengths considered (hereafter referred to as ‘profiles’), individual frequencies were expressed as deviations from the mean of all six species (i.e., Fijk−F¯jk). Since higher primates have rather similar generation lengths (Langergraber et al., 2012) and for general comparison we repeated this analysis on a second group, the Afrotheria, which have an exceptionally wide range of generation lengths, from ∼2 years for the Cape elephant shrew up to 25 years for the African elephant (Pacifici et al., 2013).

The resulting profiles for primates reveal obvious and rather complicated differences between species (Figs. 1 and 2) that increase in amplitude with repeat number, probably reflecting the higher mutation rates of longer microsatellites (Kelkar et al., 2008), and appear more extreme for AT motifs where the rate of slippage is higher (Schlötterer & Tautz, 1992). Even profiles from the closest relatives, the bonobo (blue) and chimpanzee (red), are easily distinguished, implying that shape changes can occur over timescales as short as a million years. Large profile differences might arise from differences between sequencing platforms, depth of coverage and state of assembly. For example, long microsatellites might be rare in genomes assembled from short reads. However, this is not what we see. Species with few long microsatellites at one motif usually show excesses either of long microsatellites of the other motif or of medium long microsatellites of the same motif. More generally, most profiles recall pulses moving through the length-frequency distribution, with excesses at one length balanced by deficits at another and, for any given species, the peaks and troughs tending to occur at different repeat numbers in the two motifs shown. Profiles for the Afrotheria (Figs. 3 and 4) show broadly similar features, though with generally greater amplitude due to the larger evolutionary distances between species. As with the primates, rank order of species changes constantly with repeat number and profiles for a given species often differ dramatically between AT and AC.

Figure 2 Variation in relative frequency of different length AT microsatellites in higher primates.

For details, see the legend for Fig. 1.

Figure 3 Variation in relative frequency of different length AC microsatellites in seven Afrotheria.

The seven Afrotheria are: hyrax (PCA, green), tenrec (ETE, black), manatee (TMA, light blue), elephant shrew (EED, pink), golden mole (CAS, blue), elephant (LAF, red) and aardvark (OAF, yellow). All frequencies are expressed as natural logarithms and individual data points are then calculated as deviations from the group mean. A value of one thus indicates a frequency 2.7 times the geometric mean of the seven species.

Figure 4 Variation in relative frequency of different length AT microsatellites in seven Afrotheria.

For details, see the legend for Fig. 3.

(b) Do microsatellite frequency differences correlate with phenotype?

To test for links to life history traits we fitted multiple regressions with phylogenetic correction (see Methods). Associations are summarised as estimates of the slope, to give the size and direction of any relationship, with associated confidence intervals to allow significance to be judged (Figs. 5 and 6, for raw data see Table S2, for triplet repeat plots see Figs. S4 and S5). For comparison we also present the same analysis for microsatellites carrying one interruption (Figs. 7 and 8). The plots for interrupted tracts are generally very similar to those of equivalent pure tracts, presumably because most interrupted tracts derive rather recently from a pure tract of very similar length. Supporting previous observations that microsatellite mutation rate increases with repeat number, trend strength tends to increase with repeat number (Weber, 1990; Kelkar et al., 2008; Sun et al., 2012). A general lack of significant trends among the shortest microsatellites could reflect either the lack of causal link or the mutation rate of short microsatellites, which may be too low to track phenotype evolution in mammals. Note, in interpreting these graphs it is important not to forget the temporal element. The slope estimates are based on PICs and therefore on average tend to reflect how short-term changes in phenotype covary with short-term changes in microsatellite frequency. Over longer periods the observed trends may be different and possibly even reverse.

Figure 5 Relationship between body temperature and microsatellite frequency.

Three microsatellite motifs were analysed (AT, AC and AG = black, grey and white data series respectively). The vertical axis is the estimate of the slope of the relationship between body temperature and loge microsatellite frequency, corrected for body mass and the genome’s GC%. All variables are included as phylogenetically independent contrasts. Three data points with high leverage were omitted. Error bars are one standard error of the estimate.

Figure 6 Relationship between body mass and microsatellite frequency.

Three microsatellite motifs were analysed (AT, AC and AG = black, grey and white data series respectively). The vertical axis is the estimate of the slope of the relationship between body mass and loge microsatellite frequency, corrected for body temperature and the genome’s GC%. All variables are included as phylogenetically independent contrasts. Three data points with high leverage, all corresponding to deeper nodes, were omitted. Error bars are one standard error of the estimate.

Figure 7 Relationship between body temperature and the frequency in interrupted microsatellites.

The analysis presented in Fig. 5 is repeated, but this time using microsatellites carrying a single interruption (for full definition see text).

Figure 8 Relationship between body mass and the frequency in interrupted microsatellites.

The analysis presented in Fig. 6 is repeated, but this time using microsatellites carrying a single interruption (for full definition see text).

Overall significance of the trends is difficult to assess due to the need to control jointly for both 162 multiple tests (= 27 lengths × 3 motifs × 2 phenotypes) and some unknown level of autocorrelation between similar length microsatellites. Nonetheless, there are two reasons to believe the trends are genuine. First, many individual slopes differ significantly either from zero or from each other. Second, the three motifs all show very similarly shaped profiles for each phenotype, while the two phenotypes yield approximately inverse patterns, as indicated by the negative correlation between the slopes for mass and temperature (Figs. S6–S8). Thus, the effect of increasing body temperature tends to be similar to that of decreasing body size and hence increasing population size, consistent with higher mutation rates in larger populations.

(c) Can phenotype be predicted from the frequencies of different length microsatellites?

As indicated, overall significance is difficult to determine due to non-independence. Moreover, the assumption of Brownian evolution that underpins the PIC analysis is unlikely to be met. Consequently, we sought an overall view of the informativeness of these profiles through a form of cross-validation. Each species was assigned to one of 13 major clades, each separated from all other species by at least 50 million years: Artiodactyls (n = 8); Cetaceans (n = 2); Perissodactyls (n = 2); Chiroptera (n = 6); Carnivores (n = 6); Eulipotyphla (n = 3); Primates (n = 14); Glires (n = 14); Tree shrews (n = 1); Afrotheria (n = 7); Xenarthra (n = 2); Marsupials (n = 3); Monotremes (n = 1). Note, in view of the unusual ecology of Cetaceans, Cetartiodactyls were divided into the two antiquated constituent groupings.

Body mass and body temperature were then estimated for each species in turn, using simple linear regressions with no phylogenetic correction, based only on data from the 12 other clades. Specifically, multiple regressions were constructed with the phenotype of interest as the response variable with predictor variables microsatellite frequency, F, defined above and the other phenotype. Regressions were fitted for all combinations of repeat type (N = 3) and repeat number (range 4–30). Wherever a significant regression (P ≤ 0.01, for other thresholds see Table S3) was obtained, the R function ‘predict’ was used to obtain an estimate of the response phenotype for the focal taxon. Non-significant regressions were not used since these will largely contribute only noise. Predicted phenotypes were then taken as the average of all individual estimates across all repeat numbers and repeat types. Microsatellite length frequency profiles have significant predictive power for both body temperature (r2 = 12%, N = 68, P = 0.004) and body mass (Fig. 9, r2 = 38%, N = 68, P = 1.5 × 10−8, r2 rising to 43% if the killer whale is excluded). Given that estimates of mass vary between databases, particularly for strongly sexually dimorphic species, we tested the robustness of our analysis repeating it twice, each time randomly either doubling or halving each species’ mass. The resulting r-squared values of 41% and 36% indicate that the six orders of magnitude of mass covered by our species range completely dominate any biologically plausible variation in individual species estimates.

Figure 9 Using microsatellites to predict body weight.

All data are expressed as log10 (mass in kilograms). The horizontal axis is actual adult body weight. The vertical axis is body weight predicted from the frequencies of microsatellites in that species’ genome, using simple multiple regressions based on all species not in the same major clade. The arrow indicates the killer whale, a large outlier.

Discussion

Here we test the hypothesis that the widely reported link between substitution rate and life history traits extends to neutral, rapidly evolving microsatellites. We find that the frequencies of different length microsatellites vary greatly even between closely related species, and that these differences correlate with both body temperature and body mass. Our results support a link between evolutionary rate and life history traits but the patterns we find are difficult to explain by a simple model where larger species evolve more slowly due to longer generation length.

The complicated patterns of frequency differences seen in Fig. 1 show that models based on mutation number are inadequate. A key feature of mutation number models is that rank order frequency is expected to be constant (or the inverse) across all motif-length combinations: a species’ rank depending only on the number of generations they have been through since the common ancestor of mammals. Constant rank order is also expected if species differences arise mainly due to differences between sequencing platforms, with some species carrying fewer long microsatellites across all motifs. That rank order varies greatly (54 different orders for the six primates in 81 length-motif combinations) therefore suggests no more than a minor role for generation length and argues against large sequencing platform biases. Similar results are found for the Afrotheria. The relatively minor impact of sequencing platform and assembly state is also indicated by the correlations we find between microsatellite frequency and phenotype. Indeed, since differences due to sequencing platform almost certainly exist and would inflate the error variance, our reported results most likely underestimate the true effect sizes.

Apart from generation length, three other models have been proposed for how mass and substitution rate might be correlated (Bromham, 2011). First, mass might exert an influence through a correlation with metabolic rate and/or body temperature. However, we show an independent effect of body mass in models where body temperature is also fitted. Moreover, body mass and body temperature are only weakly correlated (r2 = 0.031, n = 596, data from Clarke & Rothery (2008)). Second, mutation rate should evolve to increase with a species’ need to gamble (Giraud et al., 2001). Larger species produce fewer, individually costlier offspring so may favour less gambling and therefore lower mutation rates. This explanation fails because microsatellites are mostly neutral and evolve primarily through slippage (Schlötterer & Tautz, 1992), so microsatellites are unlikely to be affected by evolved changes in substitution rate, should these exist and evolve at a high enough rate in higher vertebrates. Microsatellite mutation rates might vary rapidly through evolved changes in the mismatch repair system, particularly if a selfish evolutionary arms race were involved. However, this model fails through lack of any plausible link to phenotype: selfish processes tend to be ‘blind’.

The last possibility relates to smaller species tending to have both higher fecundity and larger population sizes (Damuth, 1981). Genetic drift operates more slowly in large populations, thereby allowing a higher proportion of mildly deleterious mutations to be removed by selection (Ohta, 1987). This model also founders on two key elements. Most importantly, the nearly neutral theory predicts slower evolution in larger populations, the exact opposite of what has been observed (Amos, 2011). Our current data agree: in Fig. 2 the slopes for the two phenotypes are generally negatively correlated (Fig. S3), suggesting that an increase in body temperature (= more slippage) has a similar impact to a decrease in body size. There is also the recurrent problem with models that invoke selection that, while some microsatellites occur in functional regions and may be selected (The Huntington’s Disease Collaborative Research Group, 1993), most are likely neutral (Buschiazzo & Gemmell, 2010) or effectively so.

One model not previously considered for how microsatellite frequencies and body mass might become correlated is heterozygote instability (HI). The HI hypothesis proposes that recognition and ‘repair’ of heterozygous sites during meiosis leads to additional rounds of DNA replication that in turn provide opportunities for extra mutations (Amos, 2010b). HI predicts that mutation rate should scale with population size and hence with body mass due to the classical positive relationship between heterozygosity and population size (Hartl, 1988; Slatkin, 1995). Evidence for HI comes variously from the way SNPs are clustered, which fits well with a model where new mutations occur preferentially near pre-existing polymorphisms (Drake, 2007; Amos, 2010a; Schrider, Hourmozdi & Hahn, 2011), from the reduction in mutation rate that occurred as humans lost variability leaving Africa (Amos, 2013) and, most directly, from the correlation between microsatellite mutation rate and modern human population size (Amos, 2011). Direct mutation counting gives conflicting results, two studies providing support for HI (Amos et al., 1996; Masters et al., 2011) but another not (Sun et al., 2012), though the latter appears not to control for the greater detectability of mutations in homozygotes.

While HI remains an unproven hypothesis, it has the potential to explain some aspects of our results that cannot be accommodated easily by existing models. First, HI acts directly on mutation rate rather than mutation number, so could change the shape of a microsatellite length-frequency profile. Second, HI can explain mutation rate differences between morphologically and physiologically similar relatives such as chimpanzees and bonobos, as long as they differ in their demographic histories. Third, HI can account for links to life history traits for both base substitutions (Amos, 2013) and slippage in microsatellites (Amos, 2011), and operates independently of natural selection. Finally, the impact of body mass is consistent with predictions from HI because, as discussed above, higher rates of slippage seem linked to greater population size (= smaller mass). Despite this, HI does not offer a complete explanation because it cannot account for links between life history traits and haploid regions of the genome such as mitochondrial DNA (Samuels, 2004; Khaidakov, Siegel & Shmookler Reis, 2006; Welch, Bininda-Emonds & Bromham, 2008; Nabholz, Glémin & Galtier, 2009).

Comparing the ways microsatellite frequencies correlate with body mass and body temperature reveals an apparent contradiction. While the largest and most significant effect sizes are found for body temperature, the greatest predictive power is for body mass. The answer probably lies with the timescales over which the two traits vary. Body temperature can go up as well as down and may well change rapidly, for example if a species evolves hibernation. Such patterns are captured effectively by independent contrasts, a method that focuses on trait changes relative to a (usually recent) common ancestor. By comparison, body mass tends to be much more stable, varying much more between than within major clades and most trends are towards larger body size (Alroy, 1998). For mass, therefore, independent contrasts often embrace rather little change and tend to be relatively uninformative, while regressions across all taxa cover a wide range of sizes and are robust enough to give excellent predictability.

Mutation bias seems an integral feature of microsatellite evolution (Primmer et al., 1996; Sun et al., 2012) and may obey complicated rules (Ellegren, 2000; Xu et al., 2000). Our results also support the existence of biases. Without bias, the shape of the allele length-frequency profile will be stable because the average change in length of an allele over time is zero, regardless of variation in either mutation rate or mutation number. Profile shape changes require the joint action of mutation bias, to create asymmetry, and a change to the relative mutability of different length alleles. For example, suppose short microsatellites tend to expand under an upward mutation bias (Xu et al., 2000) and that a change to the mutation process either creates more short microsatellites or increases their relative mutability. The result would be a pulse of excess frequency that would migrate upwards in length until lost at the upper boundary. Such a model is speculative, and more modelling is needed to establish its viability, but it seems intuitively in keeping with the patterns seen in Figs. 1–4.

The ability to predict a species’ body mass to about an order of magnitude may appear unremarkable but we argue that this predictability can be seen as surprisingly good. First, mass is unlikely to influence microsatellite frequencies directly but instead must act through a (likely weak) correlation with some other factor such as population size. Second, even the high mutation rates of microsatellites are unlikely to prevent appreciable lag between the shape of the length-frequency distribution and changes to phenotype. Third, we consider only two factors in our prediction model, body mass and body temperature. If other factors are important, their omission will degrade predictive power. All these elements will inflate the error variance and suggest that more detailed modelling could allow an appreciable increase in predictability.

In conclusion, the relative frequencies of microsatellites of different lengths vary markedly between different species of mammal. We show how these differences are not random but instead correlate with two phenotypic traits, both of which plausibly influence microsatellite mutation rate. Even with crude measures of phenotype, the length frequency profiles contain enough information to make meaningful inferences about an unknown species. Although of little use on its own, the implication is that future research aimed at understanding and modelling the impact of genome-wide changes in mutation rate has the potential to open a window on aspects of historical ecology that were previously beyond our reach.

Supplemental Information

Table S1 Mammalian genomes used in this study

Columns are as labelled apart from ‘Clade’ which is the major grouping used for phenotype prediction. Given their radically different lifestyle, we separate Cetaceans from Cetartiodactyla, though results are essentially identical if this separation is not made. Columns ‘body temperature’ and ‘log10 (Mass), Kg’ give the phenotype estimates used in the current study. Sources are coded by number and listed to the right of the data.

Click here for additional data file.

Table S2 Microsatellite counts for three dinucleotide and three trinucleotide motifs

Data are presented for 69 species of mammal for AT, AC and AG, and for ATT, AAC and AGG. Counts are given for the range four to 30 repeats, expressed as log10 ((counts + 1)/number of bases assayed). GC content is given for each species in the top panel.

Click here for additional data file.

Table S3 Predicting body mass and body temperature from microsatellite frequencies

Predictions are based on regressions between log(microsatellite frequency) and phenotype. Only regressions based on motif-repeat number combinations that yield a rank correlation significant at the stated alpha level are used for prediction. This Table lists a range of different thresholds along with the resulting correlations between actual and predicted body mass and body temperature. A threshold of alpha = 1 is where data from all regressions are used. In general, the more stringent the criteria used for inclusion the stronger the correlation between actual and predicted phenotype. Results are presented for both pure repeat tracts and those containing a single interruption.

Click here for additional data file.

Figure S1 Example C++ script for counting microsatellites in genome sequence data

The program takes one character at a time sequentially from the genome file. It is assumed that the overwhelming majority of characters are the bases A, C, G, T and N in either upper or lower case. All characters with codes below 65 (=‘A’) or above 122 (‘z’) are ignored. The script can be used by pasting into a C++ compiler and editing the path where the genome files are located.

Click here for additional data file.

Figure S2 Example diagnostic plots for adequacy of fit of general linear models

The four plots were obtain using the R command “plot(model)” and are typical for models where, in particular, the major outlier node separating primates and rodents has been removed.

Click here for additional data file.

Figure S3 Mammalian phylogeny used in this study

The phylogeny is presented in Nexus format and can be drawn using any standard tree-drawing package (e.g., TreeView, 1. Page R.D.M. 1996 TreeView: an application to display phylogenetic trees on personal computers. Computer Applications in Biosciences 12, 357–358.).

Click here for additional data file.

Figure S4 Relationship between body temperature and trinucleotide microsatellite frequency

Three trinucleotide microsatellite motifs were analysed (AAC, AGG and ATT = black, grey and white data series respectively). The vertical axis is the estimate of the slope of the relationship between body temperature and loge microsatellite frequency, corrected for body mass and the genome’s GC%. All variables are included as phylogenetically independent contrasts. One major outlier was removed. Error bars are one standard error of the estimate.

Click here for additional data file.

Figure S5 Relationship between body mass and trinucleotide microsatellite frequency

Three trinucleotide microsatellite motifs were analysed (AAC, AGG and ATT = black, grey and white data series respectively). The vertical axis is the estimate of the slope of the relationship between body mass and loge microsatellite frequency, corrected for body temperature and the genome’s GC%. All variables are included as phylogenetically independent contrasts. One major outlier was removed. Error bars are one standard error of the estimate.

Click here for additional data file.

Figure S6 Correlation between the relationships between body mass, body temperature and relative AC microsatellite frequency

The two axes correspond to the estimates of the two phenotypes taken from Figs. 5 and 6 and show a broadly inverse relationship.

Click here for additional data file.

Figure S7 Correlation between the relationships between body mass, body temperature and relative AG microsatellite frequency

The two axes correspond to the estimates of the two phenotypes taken from Figs. 5 and 6 and show a broadly inverse relationship.

Click here for additional data file.

Figure S8 Correlation between the relationships between body mass, body temperature and relative AT microsatellite frequency

The two axes correspond to the estimates of the two phenotypes taken from Figs. 5 and 6 and show a strongly inverse relationship.

Click here for additional data file.

Additional Information and Declarations

Competing Interests

Author Contributions

William Amos is an Academic Editor for PeerJ.

William Amos conceived and designed the experiments, performed the experiments, analyzed the data, contributed reagents/materials/analysis tools, wrote the paper, prepared figures and/or tables, reviewed drafts of the paper.

Laura N.S. Filipe performed the experiments, wrote the paper, reviewed drafts of the paper.

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
