# Peer review of "Microsatellite frequencies vary with body mass and body temperature in mammals, suggesting correlated variation in mutation rate"

_PeerJ, doi:10.7717/peerj.663_

## Round 0.1 · original submission · Major Revisions

The reviewers suggest a number of minor revisions to the work, but also point out some larger concerns. I found particularly relevant reviewer #1's concerns about prediction, reviewer #2's points about mass estimates and possible inclusion of generation time, and reviewer #3's argument about substitution rates splitting tracks, time estimates on the tree, and reviewer #1 and #3's thoughts on the justification and evidence against a mutation number model. I had a few minor additional points myself:

Line 79: In the spirit of reproducibility, it would be very nice to see the custom C++ script made available online in a repository like github/bitbucket/figshare etc, so that others have easy access.
Line 82: I'm perplexed by leaving out GC microsatellites. A quick BLAST of the mouse genome with a GC(10) repeat finds ~300 perfect matches. Since there appear to be differences among other classes as well, it is not clear to me why these should be excluded.
Line 158: the second "trends" should read "tends"
Line 248: "considered for how microsatellite"

Reviewer 1 ·

Basic reporting

Amos and Felipe investigate the relationship between microsatellite mutation and two traits, body mass and body temperature, across 69 mammalian species using publicly available genome sequences. They demonstrate that generation length alone cannot explain the observed differences in the microsatellite length distribtuion. They call upon the correlation between body size and population size to suggest that heterozygote instability could be the underlying cause linking body size and mutation rate. There are a few instances, indicated below and in the following sections, where the data do not convincingly demonstrate the conclusions that are drawn.

The article appears to meet most of the basic reporting standards, with a few exceptions:

1. The sentence from line 45-47: “However, regardless of the exact details, the relative frequency of any given length of microsatellite in the genome should tend to reflect the relative time spent by an average microsatellite during its lifetime at that length”, could benefit from either a reference or a more complete explanation.

2. The sentences from line 61-62: “Changes in mutation number should affect all repeat numbers equally. In contrast, changes in mutation rate may affect longer and shorter microsatellites differentially.” Should have a reference or a more complete justification, especially since this sentence is fundamental to the study.

3. The sentence on lines 88-89: Since a minority of tracts are interrupted and interrupted tracts seem to mutate approximately as if they comprise the longest pure tract...”, should be justified with data or a reference. It appears that this is an observation from the data, in which case demonstrating it with the data would be ideal.

4. The “classical correlation between heterozygosity and populatio size” should be cited (lines 251-253).

Experimental design

It was nice to see that the authors investigated model fit (lines 102-103).

1. A pivotal conclusion, that microsatellite frequency differences cannot be explained with a mutation number model, is based only on data from higher primates. Given the importance of this finding, and that the authors generalize it to all mammals, they should conduct the analysis for at least one other set of closely related species, which presumably their data contain.

2. It is disturbing that although the trends relating phenotype and microsatellite frequency were insignificant, after accounting for multiple testing, the authors suggest that insignificance should be ignored (lines 167-170). Perhaps an FDR or other type of evaluation could be used to establish significance, if it really is there. For instance, one approach might be for the authors to look for enrichment of significant trends at the 5% level (i.e. are many more than 5% of the tests significant at the 5% level?).

3. The authors attempted to overcome the lack of significance (above) by predicting phenotypes from microsatellite length frequency profiles. Yet, they excluded all non-significant regressions “as these will largely contribute only noise”. That is exactly the point—how predictive are the data including the noise? It appears that this exclusion may lead to a substantial bias toward false-positive predictive power. Predictive ability when insignificant regressions are included should be reported.

Validity of the findings

The validity of the findings is difficult to assess until the above statistical issues are addressed.

·

Basic reporting

This is an interesting and well written article. The resolution of the figures are however clearly too low. They appear blurred in the pdf.

Minor comments:

Line 97: The package “ape” should be cited using Paradis et al. (2004) reference.
Line 162: The sentence is not strictly correct as PIC could also reflect long-term changes when contrasts are made between long-branches so it ultimately depend on the data set analyzed. It would be better to say that PICs involved, on average, short-terms changes.
Line 241: At the end of this sentence a reference is made with a number “[43]”.

Reference:
Paradis E, Claude J, Strimmer K (2004) APE: Analyses of Phylogenetics and Evolution in R language. Bioinformatics, 20, 289–290.

Experimental design

- The authors estimates 27 length x 3 motifs = 81 microsatellite frequencies per species. Then, these estimates are all used in correlation with body-mass and temperature leading to a lot of correlations. A way to simplify the analysis could be to use multivariate analysis such as principal component analysis to reduce the number of explanatory variables and then, use principal components as covariates with life-history traits (see principal component regression method). Alternatively, the authors could also merge several length categories together, for example, merge all the microsatellite of one particular motifs with a length >10 repeats.

- I quickly compare the body-mass used in the present study with values obtained from two classical databases, namely PanTHERIA (Jones et al. 2009) and AnAge (Tacutu et al. 2012). Although the values are well correlated, I do found some discrepancies. For example, Amos & Filipe use 10^(-1.3372422) = 46g for house mouse whereas AnAge reports 20.2g and PanTHERIA, 19.3g, similarly, for the domestic goat (Capra hircus), the authors reports 10^(1.255)=17.9Kg whereas AnAge reports 61Kg and PANTheria 46.9Kg. This is unlikely to radically change the results but I recommend that the authors reproduce their correlation using another database (AnAge?) to test the robustness of their results.

References:
Jones KE, Bielby J, Cardillo M et al. (2009) PanTHERIA: a species-level database of life history, ecology, and geography of extant and recently extinct mammals (WK Michener, Ed,). Ecology, 90, 2648–2648.
Tacutu R, Craig T, Budovsky A et al. (2012) Human ageing genomic resources: integrated databases and tools for the biology and genetics of ageing. Nucleic acids research, gks1155.

Validity of the findings

No Comments

Additional comments

A large body of literature exists linking mitochondrial repeats number and species longevity in mammals (Samuels 2004; Khaidakov et al. 2006; Guo et al. 2010; Lakshmanan et al. 2012). There is a negative correlation between the number of repeats and species longevity. The rationale is that the number of repeats is constrained in long-lived species to avoid somatic deletion of the mitochondrial genome ultimately involved in the ageing process. Because nuclear DNA mutation could also be involve in ageing (Best 2009), the present result clearly echoes the results obtained with mitochondrial genome. Therefore, I think that it is important that the authors (i) make reference to the previous results obtained with the mitochondrial genome and (ii) use maximum longevity as an additional response variable in their analyzes. Maximum longevity could be easily obtained using AnAge database (http://genomics.senescence.info/species/ Tacutu et al. 2012).

References
Best BP (2009) Nuclear DNA damage as a direct cause of aging. Rejuvenation research, 12, 199–208.
Guo X, Popadin KY, Markuzon N et al. (2010) Repeats, longevity and the sources of mtDNA deletions: evidence from “deletional spectra.” Trends in Genetics, 26, 340–343.
Khaidakov M, Siegel ER, Shmookler Reis RJ (2006) Direct repeats in mitochondrial DNA and mammalian lifespan. Mechanisms of Ageing and Development, 127, 808–812.
Lakshmanan LN, Gruber J, Halliwell B, Gunawan R (2012) Role of direct repeat and stem-loop motifs in mtDNA deletions: cause or coincidence? PloS one, 7, e35271.
Samuels DC (2004) Mitochondrial DNA repeats constrain the life span of mammals. Trends in Genetics, 20, 226–229.
Tacutu R, Craig T, Budovsky A et al. (2012) Human ageing genomic resources: integrated databases and tools for the biology and genetics of ageing. Nucleic acids research, gks1155.

·

Basic reporting

Line 16 - The authors cite Thorpe (1982) for the 'molecular clock', which is primarily a review of various methods for estimating divergence times from molecular data; I recommend the authors instead use Zuckerkandl and Pauling (1962, Horizons in Biochemistry) and Margoliash (1963, PNAS) rather than, or in addition to, Thorpe.

Line 21 - Lartillot and Poujol (2011, MBE) -- which finds evidence for a negative correlation between substitution rates and both body size and generation time in mammals -- seems like a necessary addition to this list of studies of correlation between life-history and evolutionary rates.

Experimental design

I have major concerns with the design and conduct of the research.

Data:
The description of the pipeline for identifying microsatellites based on their motif and length is ambiguous, and the categorization of microsatellites is potentially biased. In particular, on line 87 it is stated that "Substitutions ... occur at an appreciable rate and act to reduce slippage", suggesting that nucleotide substitutions occur with some 'appreciable rate' within given microsatellites. However, the next sentence reads "... a minority of tracts are interrupted...", which seems to indicate the opposite: that substitutions within microsatellites are rare (assuming an 'interrupted tract' is one which has been interrupted by a substitution). Further, it is asserted that "... interrupted tracts seem to mutate approximately as if they comprise just the longest pure tract, [so] we counted only pure repeat tracts". No evidence or citation is provided for this assertion -- if the authors performed analyses to determine that this was not a significant problem, those results should be presented as supplemental material. Additionally, line 90 describes categorizing an (AC)_10AT(AC)_4 sequence as one (AC)_10 tract and a separate (AC)_4 tract. It is not clear how such a sequence is distinguished from an (AC)_15 sequence with a single substitution, which would place it in the length-15 bin, and it is also seems to be at odds with the use of 'only pure repeat tracts', which would place it in the length-10 bin (without the remainder being placed in the length-4 bin, if my interpretation is correct).
These issues are of great concern because longer microsatellites are larger targets for substitution, and thus we might expect larger microsatellites to be subdivided at a higher rate, leading to an excess of mid-sized microsatellites, according to the presented procedure. A correlation between body size and substitution rate, which is a well-established pattern in the mammals (and is noted by the authors, line 22), is thus a potentially major confounding factor in the analyses as presented.
Describing a complete probabilistic model of microsatellite birth, evolution, and death is clearly beyond the scope of this work. However, at the very least it is necessary to attempt to mitigate the effects of substitutions within microsatellites, for example by connecting tracts of the same motif that are separated by relatively small gaps.

Statistical Analysis:
The ages of nodes in the tree provided as supplemental material are quite different from those in Meredith et al. 2011. The authors claim to use node ages read directly from the Meredith et al. tree whenever possible, and exclusively for all 'deeper nodes'. However, the crown age of the tree provided by the authors is 126 mya, well outside the credible interval estimated for crown Mammalia by Meredith et al., (203.3-238.2 mya). Similarly, crown Theria are placed at 58.5 mya by the authors, and between (167.2-215.3) by Meredith et al. This underestimation may be consistent, but the recoding of taxon labels in the provided supplemental tree makes it challenging for the non-mammologist to identify nodes shared with the Meredith et al. tree. If these differences are the result of a simple rescaling of the estimated node ages according to Meredith et al., then this poses no problem: the comparative analyses used are blind to scalar transformations of time. However, because no rescaling is discussed, and since additional clades from other sources have been added to the analysis, it is worrying that node ages are inconsistent with the Meredith et al. tree, and all analyses performed using this tree are suspect. Node ages should be checked for consistency with both the Meredith et al. tree where applicable and with original sources for any clades not originally in the Meredith et al. tree. Additionally, taxon labels need to be consistent between the Meredith et al. tree, all supplemental data, and the supplemental tree, in order to maximize transparency.
The description of PIC analyses and linear regressions is too brief. Model fit seems to have been assessed by visual inspection -- these plots should be presented as supplemental material. It is unclear whether the authors performed any formal model comparison (using e.g. likelihood ratio tests or AIC) between linear models with and without different predictor variables to assess relative model performance, as is common practice. The significance of a particular predictor should be assessed by comparison of models variously including and excluding it, rather than by estimating its effect to be non-zero, as seems to be done here. The use of %GC content as a predictor is well-motived, but it is not obvious that it is treated appropriately in the PIC analysis: in particular, PIC assumes a trait evolves under Brownian motion and can take on any real value, whereas a proportion is constrained to the interval [0,1]. This same problem motivates the use of log-transformed body mass values, however in the case of %GC a logit-transform seems more appropriate.

Validity of the findings

The authors present evidence from Hominoids that microsatellite-length frequency profiles vary between species in a way that cannot be consistent with simple interspecific variation in generation time (what they term 'mutation number models'). They conclude from variation among species in rank order across frequency classes that number models are inadequate: number models predict that the expected rank order is the same across frequency classes (as seen in Figure 1). However, the strength of this result is difficult to quantify. Undoubtedly the process of microsatellite birth, evolution, and death has a (potentially large?) stochastic element and thus our expectation (rank-order constancy) is but one of many possible outcomes; however, without a rigorous statistical model, it is hard to determine what one should 'expect' these rank order distributions to look like, and how surprised one should be by the ones observed. Perhaps more problematical, the species analyzed (all members of hominoidea) have similar generation times: the gibbon probably has the shortest generation time at about 15 years (Chan et al. 2013 'Inferring the evolutionary histories of divergences in Hylobates and Nomascus gibbons through multilocus sequence data', BMC Evol Biol), whereas the rest of the species have remarkably similar generation times of about 27 years (Langergrabera et al. 2012 'Generation times in wild chimpanzees and gorillas suggest earlier divergence times in great ape and human evolution', PNAS). This range of generation times is both small and likely unrepresentative of all of the mammals, whose generation times range from 3 months to 50 years (Pacifici et al. 2013 'Generation length of mammals', Nature Conservation 5). The data and analyses as presented are inadequate to convincingly rule out an effect of generation time on the evolution of microsatellites. More data are needed, and some way of judging the 'surprisingness' of the data is called for (see below).
The analysis of correlation between phenotype and frequency differences is suggestive of a real pattern, but falls short of being entirely convincing. Primarily, concerns about how the data were curated and analyzed (see data and statistical analysis, above) make it difficult to take the patterns at face value. Additionally, as the authors are aware of and are particularly open about, the nature of the data and the inappropriateness of the model used to analyze them obscure the statistical significance of these results. The Brownian motion model implicit in the PIC analyses assumes that the frequencies of each motif-length combination evolve independently of each other: however, it is clear that microsatellites evolve between frequency classes of the same motif (as the authors note, frequency classes are 'autocorrelated'), so the independence assumption is violated in a way that is potentially unquantifiable. There is also another source of non-independence that the authors allude to in a different context -- microsatellite frequencies across motifs may be non-independent, as indicated on line 146: "Species with few long microsatellites at one motif usually show excesses ... of long microsatellites of the other motif ...". These sources of non-independence are likely to artificially increase certainty in estimates of effect sizes (i.e., shrink error bars), but the severity of the problem is unclear.
Both of the afore-mentioned concerns stem from the absence of a clearly defined and tractable model for microsatellite evolution. Barring the development of such a model, it would at least be necessary to perform a small simulation study of a 'realistic' mechanistic process of microsatellite evolution (including birth, evolution, and death) to understand the effects of autocorrelation and non-independence on the methods used in this manuscript, and could provide useful a null model for frequency rank-order distributions. This would help develop the reader's intuition about the evolutionary behavior of microsatellites and would provide a critical baseline against which to compare the present findings.

---

## Round 0.2 · accepted · Accept

Many thanks for your careful consideration of the referees' suggestions. Given these revisions, I see no reason the manuscript should not be publishable in PeerJ. Regarding the remaining outstanding questions: please do upload your C++ code (supplemental material would be fine), and attaching a set of four "typical plots" as suggested would certainly be acceptable.